# Derivation of Oligodendrocyte Precursors from Adult Bone Marrow Stromal Cells for Remyelination Therapy

**DOI:** 10.3390/cells10082166

**Published:** 2021-08-22

**Authors:** Yat-Ping Tsui, Guy Lam, Kenneth Lap-Kei Wu, Maximilian Tak-Sui Li, Kin-Wai Tam, Daisy Kwok-Yan Shum, Ying-Shing Chan

**Affiliations:** 1School of Biomedical Sciences, LKS Faculty of Medicine, The University of Hong Kong, Hong Kong, China; u3506833@connect.hku.hk (G.L.); kenneth_wu@hku.hk (K.L.-K.W.); tslimax@hku.hk (M.T.-S.L.); anthonykwtam@connect.hku.hk (K.-W.T.); shumdkhk@hku.hk (D.K.-Y.S.); 2State Key Laboratory of Brain and Cognitive Sciences, The University of Hong Kong, Hong Kong, China

**Keywords:** cell therapy, oligodendrocyte precursors, bone marrow stromal cells, directed differentiation, myelin disorders

## Abstract

Transplantation of oligodendrocyte precursors (OPs) is potentially therapeutic for myelin disorders but a safe and accessible cell source remains to be identified. Here we report a two-step protocol for derivation of highly enriched populations of OPs from bone marrow stromal cells of young adult rats (aMSCs). Neural progenitors among the aMSCs were expanded in non-adherent sphere-forming cultures and subsequently directed along the OP lineage with the use of glial-inducing growth factors. Immunocytochemical and flow cytometric analyses of these cells confirmed OP-like expression of Olig2, PDGFRα, NG2, and Sox10. OPs so derived formed compact myelin both in vitro, as in co-culture with purified neurons, and in vivo, following transplantation into the corpus callosum of neonatal shiverer mice. Not only did the density of myelinated axons in the corpus callosum of recipient shiverer mice reach levels comparable to those in age-matched wild-type mice, but the mean lifespan of recipient shiverer mice also far exceeded those of non-recipient shiverer mice. Our results thus promise progress in harnessing the OP-generating potential of aMSCs towards cell therapy for myelin disorders.

## 1. Introduction

White matter diseases encompass vast groups of demyelinating, dysmyelinating, and hypomyelinating disorders which often result in severe loss of motor and cognitive functions, and even death [1,2,3]. Dysfunction or loss of myelinating oligodendrocytes (OLs) is commonly observed in the pathogenesis of such disorders. Transplantation of oligodendrocyte precursors (OPs) for maturation into myelinating OLs in replacement of lost or dysfunctional OLs thus holds great promise for remyelination therapy and related recovery of function [4,5,6,7].

Numerous Phase I trials for transplantation of stem cells and neuroprogenitor cells into the human brain demonstrate the feasibility of cell replenishment therapy for neurological disorder [8,9,10]. However, transplantation of multipotent stem cells, such as embryonic stem cells and induced pluripotent stem cells (iPSCs), incurs a much greater risk of teratoma formation compared to lineage-restricted progenitors. Moreover, transplantation of embryonic stem cells raises ethical concerns [11,12], while the derivation of OPs from iPSCs currently involves multi-step procedures that require more than 100 days [13,14,15]. We envision that derivation of OPs from neuroprogenitor cells harbored in bone marrow stroma overcomes these hurdles and paves the way for clinical implementation.

Derivation of myelin-forming Schwann cells from adult bone marrow stromal cells (aMSCs) showed that neural progenitors were harbored in aMSCs [16,17,18,19,20,21]. We hypothesized that these neural progenitors could be expanded and differentiated into OPs, without genetic manipulation. In this study, we aim to test if aMSC-derived OPs can mature into myelinating OLs in myelin-deficient shiverer mice.

The ability to derive OPs from cryopreserved aMSC stocks within a short time frame highlights bone marrow banks as a resource that can be tapped into. This facilitates donor matching and shortens the time that patients spend waiting for treatment. Neural progenitors can maintain marker expression profiles despite cryopreservation [22,23], but less is known about the potential to differentiate into OPs following reconstitution. As such, we specifically used aMSCs reconstituted from preservation under liquid nitrogen for the production of OPs as described in this study.

Here we report successful generation of Olig2, NG2, PDGFRα, and Sox10-expressing OPs from cryopreserved aMSCs with the use of a 2-step protocol in 21 days. These OPs were capable of forming compact myelin in vitro and, more importantly, in vivo following intracerebral delivery into myelin-deficient shiverer mice. Remyelination observed in the corpus callosum of shiverer mice was accompanied by behavioral rescue from autonomic deficits and extended lifespan. These results give promise to the sourcing of adult bone marrow for neural progenitors that can be expanded and differentiated into OPs for remyelination therapy.

## 2. Materials and Methods

### 2.1. Animals

Young adult Sprague Dawley rats (200–250 g, Charles River Laboratories, Wilmington, MA, USA) were used for bone marrow extraction. OPs were harvested from neonatal Sprague Dawley rats for comparison of marker expression with aMSC-derived OPs. Dorsal root ganglia were harvested from embryonic day 15 Sprague Dawley rats. Homozygous shiverer mice (C3Fe.SWV-MBP^Shi^/J, The Jackson Laboratory, Bar Harbor, ME, USA) were used as recipient of aMSC-derived OPs to demonstrate in vivo myelination as a result of OP transplantation into the corpus callosum. C57BI/6J mice (The Jackson Laboratory) were used to illustrate degree of myelination in the wild type. All experimental protocols were approved by the Committee for Use of Live Animals in Teaching and Research, The University of Hong Kong.

### 2.2. Bone Marrow Extraction, aMSC Culture and Cryopreservation

aMSCs were cultured and characterized according to our previous publications [16,17,18,19,20,21] with modifications. Briefly, young adult rats were euthanized with anesthetic overdose (pentobarbital 150 mg/kg i.p., Alfasan, Utrecht, The Netherlands). Femurs were then dissected and whole marrow was flushed out with PBS and collected by centrifugation (480× *g*, 5 min, 4 °C). Pellets were resuspended in a growth medium comprising MEM-alpha medium (Thermo Fisher Scientific, Waltham, MA, USA) supplemented with FBS (10%, Biosera, Kansas City, MO, USA) and maintained at 5% CO_2_, 37 °C. Unattached cells were removed by medium change 48 h later. aMSC colonies were detached with the use of TrypLE Express (Thermo Fisher Scientific) and passaged at 1:3 (*v/v*) in the growth medium.

aMSC cultures between passages 3–8 were detached with use of TrypLE Express, centrifuged (250× *g*, 5 min, 4 °C), resuspended in cryopreservation medium comprising MEM-alpha medium supplemented with 10% FBS and 10% DMSO (Sigma-Aldrich, St Louis, MO, USA) and stored in liquid nitrogen in 1.5 mL cryopreservation vials (Nunc, Roskilde, Denmark). To recover aMSCs, cryopreservation vials were warmed in a 37 °C water bath until no solid ice was observable in the vials. Contents of cryopreservation vials were diluted in the growth medium, centrifuged (250× *g*, 5 min, 24 °C), resuspended in the growth medium, and seeded at a seeding density of 30,000 cells/cm^2^.

### 2.3. aMSC Proliferation Assay

Thawed aMSCs seeded in 6-well culture plates were passaged as mentioned above. For each experiment, two 6-well plate wells were seeded with aMSCs. The time between each passage was recorded and the number of aMSC per well was recorded between passages 4–8. No statistically significant difference was observed when aMSCs from different rats were used.

### 2.4. Neural Progenitor Derivation

Neuroprogenitor cells were selectively expanded using non-adherent culture [16,17,18,19,20,21]. Briefly, aMSCs were seeded at 10,000 cells/cm^2^ density onto Ultra Low^®^ non-adherent 6-well plates (Corning, Corning, NY, USA) in sphere-forming medium (SFM) comprising DMEM/F12 (Thermo Fisher Scientific) supplemented with 2% B27 (Thermo Fisher Scientific), bFGF, and EGF (20 ng/mL, Peprotech, Rehovot, Israel). Cultures were maintained for 8–10 days with medium refreshed every 48 h. At timed intervals, selected spheres were partially dissociated via incubation in TrypLE Express for efficient cell counting and determination of proportion of cells immunopositive for the neural stem/progenitor marker nestin [24] and the OL lineage determining factor Olig2 [25,26]. 

### 2.5. Differentiation of Neural Progenitors along the OL Lineage and Cryopreservation

To obtain OPs, rat bone marrow sphere cells were plated onto 6-well plates coated with poly-D-lysine (Sigma-Aldrich) and laminin (Roche, Basel, Switzerland) in glia inducing medium (GIM) comprising MEM-alpha medium (Thermo Fisher Scientific) supplemented with bFGF (10 ng/mL), PDGF-AA (10 ng/mL), β-heregulin (100 ng/mL, Millipore), N2 (1%, Thermo Fisher Scientific), and FBS (2.5%). OPs were culture-expanded in OP maintenance medium comprising DMEM/F12 medium supplemented with N2, bFGF (20 ng/mL), and PDGF-AA (20 ng/mL). OPs were monitored for positivity of 4 key OP markers including Olig2, NG2, PDGFRα, and Sox10. OPs were cryopreserved in OP cryopreservation medium comprising 90% GIM (DMEM/F12 medium supplemented with N2, bFGF, PDGF-AA) with 10% DMSO in liquid nitrogen.

### 2.6. aMSC-Derived OP Proliferation Test

Thawed aMSC-derived OPs were seeded at 20,000 cells/cm^2^ density in 6-well plates. aMSC-derived OPs were passaged upon reaching 80% confluence. The time between each passage was recorded and the number of OPs per well was recorded for 2 consecutive passages. These OPs could be culture-expanded for at least 2 passages, indicating proper cell yield.

### 2.7. Isolation and Primary Culture of Rat OPs 

Primary cultures of rat OPs were prepared as previously described [27]. Briefly, cerebral cortices dissected from neonatal rats (pentobarbital 150 mg/kg i.p., Alfasan) were diced and digested with 0.25% Trypsin (Thermo Scientifics) at 4 °C. The digested mixture was triturated with a fire-polished glass pipette and then centrifuged (250× *g*, 4 °C). The cell pellet was resuspended in DMEM/F12 supplemented with FBS (5%) and the suspension was filtered through a 70 μm cell strainer (BD Bioscience, Franklin Lakes, NJ, USA). The filtrate containing OPs was plated onto PDL-coated 75 cm^2^ culture flasks and maintained for 10 days with medium refreshed every 72 h. After 10 days of culture, the flasks were screwed tightly and shaken on an orbital shaker at 200 r.p.m. for 16 h to release OPs into the medium. The OP-containing medium was centrifuged (250× *g*, 4 °C), resuspended in OP maintenance medium, and plated onto PDL-coated 6 cm culture dishes.

### 2.8. aMSC-Derived OP-Neuron Co-Culture

Dorsal root ganglia of embryonic day 15 rats were dissociated with TrypLE Express and collected by centrifugation (250× *g*, 5 min, 4 °C) [16,17,18,19,20,21]. Cells were resuspended in neuron maintenance medium comprising Neurobasal medium (Thermo Fisher Scientific) supplemented with 2% B27 and NGF (20 ng/mL, Millipore, Burlington, MA, USA) and then seeded at 100,000 cells/cm^2^ onto poly-d-lysine/laminin-coated plates. Endogenous Schwann cells and fibroblasts were eliminated following brief treatment with fluorodeoxyuridine and uridine (10 μM each, Sigma-Aldrich), whereas dorsal root ganglion neurons and neurite network remained adherent on the coated plates. 

aMSC-derived OPs were seeded at 10,000 cells/cm^2^ onto the axonal network of dorsal root ganglion neurons and the co-culture was maintained in myelination medium comprising Neurobasal medium supplemented with B27, NGF neutralizing antibody (5 μg/mL, Abcam, Cambridge, UK), thyroid hormone (10 ng/mL, Sigma-Aldrich), and DAPT (10 ng/mL, Sigma) for 14 days. Myelin-forming OLs and axons in co-cultures were immunostained for myelin basic protein (MBP) and neurofilament 200 (NF200) respectively. Co-cultures were also prepared for examination of compact myelin under transmission electron microscopy. aMSCs in co-culture with dorsal root ganglion neurons were prepared in parallel as controls.

### 2.9. Transplantation of aMSC-Derived OPs

OPs derived from rat bone marrow (300,000 cells in 1 μL of DMEM/F12 medium) were delivered into the right corpus callosum (bregma −1 mm, medial-lateral 1 mm, dorsal-ventral 1.5 mm) of myelin-deficient shiverer mice (anesthetized with 2% isoflurane in 250 ml/min O_2_, RWD, Shenzhen, China) on postnatal day 7 using a Hamilton syringe driven by a computer-controlled syringe pump (0.5 μL/min) to ensure optimal delivery and to prevent backflow. For shiverer mice in the control group, fresh DMEM/F12 medium was delivered instead. A total of 42 shiverer mice were used and 2 were lost during the surgery procedure. Of the remaining 40 mice that survived, 17 were used for histological analysis and 23 for lifespan analysis. Eight wild-type C57BL/6J mice were used as controls. Mice were perfused with 4% PFA at 6- and 12-weeks later and fixed brains were harvested and processed for histological examination under confocal and electron microscopy for MBP-positive OLs and compact myelin. For lifespan analysis, Kaplan-Meier analysis was used to assess and compare survival of mice that received OP transplant versus vehicle control mice.

### 2.10. Immunofluorescence

Cells in culture were fixed with 4% PFA for 10 min and washed in PBS for 3 × 5 min. Fixed cells were incubated in blocking buffer comprising PBS with 3% *v*/*v* normal goat serum (NGS, Millipore) or 1% *v*/*v* BSA (Sigma-Aldrich) and 0.1% *v*/*v* Triton X100 (Sigma-Aldrich) for 30 min. Samples were incubated with primary antibodies overnight at 4 °C (listed in Appendix A). After washing in 1xPBS thrice, cells were incubated with the appropriate fluorophore-conjugated Alexa Fluor^®^ secondary antibodies (diluted 1:400 in blocking buffer, Thermo Fisher Scientific) for 60 min, followed by 3 rounds of washing in 1xPBS. Hoechst stain (Sigma-Aldrich) was used to counterstain nuclei. Images were viewed under an Olympus IX71 inverted fluorescence microscope equipped with Olympus DP71 camera for image capture. For cell counting, 10 random fields were captured and a minimum of 300 cells were counted per experiment by investigators blinded to the grouping.

Mouse and rabbit isotype controls (1:200, Thermo Fisher Scientific) were used in control experiments (Appendix A).

### 2.11. Sample Preparation and Immunohistochemistry

At 6- and 12-weeks post-injection, mice were deeply anesthetized (pentobarbital 150 mg/kg i.p., Alfasan) before transcardial perfusion with ice-cold saline followed by 4% PFA. Brain tissue was dissected, post-fixed overnight in 4% PFA at 4 °C, and cryoprotected in 30% sucrose solution. Coronal cryosections (50 μm) containing the corpus callosum were prepared (Thermo Fischer Scientific NX50 cryostat). The sections were permeabilized and blocked with 1% Triton X-100 and 3% BSA in phosphate buffered saline for 60 min, and then incubated with primary antibodies overnight at 4 °C (listed in Appendix A) followed with the appropriate fluorophore-conjugated Alexa Fluor^®^ secondary antibodies (diluted 1:400 in blocking buffer, Thermo Fisher Scientific) for 3 h. Cell nuclei were counterstained with Hoechst 33258 (Sigma-Aldrich). The stained sections were rinse and mounted for viewing and image capture under confocal microscopy (Zeiss LSM 800).

### 2.12. Transmisison Electron Microscopy

PFA-fixed samples were further fixed with 1% osmium tetroxide, stained with 1% uranyl acetate, and were then embedded in Epon. Sagittal sections (75–90 nm thick) of the corpus callosum were picked up on formvar/carbon-coated 75 mesh Cu grids and stained for 20 s in 1:1 super-saturated uranyl acetate in acetone followed by 0.2% lead citrate. Images were viewed under a Philips CM100 transmission electron microscope and selected images were captured for ultrastructural analysis of myelin morphology. Within imaged areas of 100 μm² of the corpus callosum, myelinated axons showing clear major dense lines within compact myelin and larger than 1 μm² in cross-sectional area were counted and assessed for the myelin g-ratio, being the ratio of the inner radius to the outer radius of the myelin sheath.

### 2.13. Flow Cytometry

Cells were dissociated via incubation in TrypLE Express, fixed in 4% PFA for 5 min, and then sedimented by centrifugation (250× *g*, 5 min). The pellets were gently washed with PBS and then resuspended in a blocking buffer comprising PBS with 3% *v*/*v* NGS and 0.1% *v*/*v* Triton X100 for 30 min. The cells were incubated with selected primary antibodies (listed in Appendix A) for 2 h at 4 °C followed by the appropriate Alexa Fluor^®^ secondary antibodies (diluted 1:400 in blocking buffer, Life Technologies, Carlsbad, CA, USA) for 30 min. Immunolabelled cells were subjected to flow cytometric analysis with use of BD CantoII Analyzer. A minimum of 10,000 cells were analyzed per experiment.

Mouse and rabbit isotype controls (diluted 1:200 in blocking buffer, Life Technologies) were used in the control experiments.

### 2.14. Reverse Transcription Polymerase Chain Reaction (RT-PCR)

Total RNA was extracted using TRIzol (Thermo Fisher Scientific) according to the manufacturer’s protocol. Total RNA (5 μg) recovered from OPs in culture was used for first-strand cDNA synthesis using PrimeScript Reverse Transcriptase (Takara, Kusatsu, Japan) with Oligo-(dt) primers. PCR was performed with 200 ng cDNA, Taq DNA polymerase (NEB), and primers listed in Appendix A. Colorimetric analysis of the resulting bands was performed with use of a Biorad Gel Doc XR imaging system; results were normalized against that of the housekeeping gene GAPDH.

### 2.15. Statistical Analysis

Statistical analyses were performed using GraphPad Prism Software 8 (GraphPad Software Inc., San Diego, CA, USA) and R Studio. Data are presented as mean ± S.D. Student’s *t*-test and Welch’s *t*-test were used for statistical analysis and *p*-values < 0.05 were considered significant. 

## 3. Results

### 3.1. Characterization of Primary and Cryopreserved aMSCs

We extracted and expanded aMSCs as reported previously [16,17,18,19,20,21]. Primary aMSCs (passages 3–8) were fibroblastic in morphology (Figure 1A) and proliferated steadily throughout passage 4 to 8. Cultures reached 80% confluence every 31 h after each passage (Appendix A, *n* = 3 replicates). Over 95% of the aMSCs were immunopositive for the mesenchymal stem/progenitor cell markers CD90 (Figure 1B), CD73 (Figure 1C) and CD105 (Figure 1D). The neural stem/progenitor cell marker, nestin, was found positive in >15% of the aMSCs (Figure 1F). Notably, the hematopoietic marker CD45 was found positive in but <1% of the aMSCs (Figure 1E). The aMSCs were of high purity and devoid of contaminating cells of hematopoietic lineage. This marker expression profile was maintained in aMSCs even after recovery from cryopreservation (Appendix A).

### 3.2. Derivation of Neural Progenitors through Non-Adherent Culture

Following transferring to non-adherent culture, aMSCs spontaneously formed spheres that measure >50 μm in diameter within 6 days (Figure 2A). By day 10, 83.15% ± 8.38% of the sphere cells indicated immunopositivity for the neuroprogenitor cell marker, nestin (Figure 2B,F) in contrast to 12.42% ± 2.82% found among aMSCs maintained in parallel, adherent culture (Figure 2D,F). Incidentally, 89.96% ± 5.45% of the sphere cells were also immunopositive for glial fibrillary acidic protein (GFAP, Figure 2C,F) in contrast to 0.78% ± 0.45% among aMSCs in parallel, adherent culture (Figure 2E,F). On the contrary, immunopositivity for the OL lineage determining factor, Olig2, was found in 36.49% ± 7.92% of the sphere cells in non-adherent culture (Figure 2B,F), in contrast to only 0.59% ± 0.48% among aMSCs in adherent culture (Figure 2D,F). Expression patterns of the indicated marker genes among spheres in non-adherent culture were in corroboration with the results of RT-PCR analysis of the transcripts (Appendix A). The results suggest that the sphere-forming cultures fostered selective expansion of neuroprogenitor cells among the aMSCs and that an Olig2-positive subset of the neuroprogenitor cells is poised for fate restriction to the OL lineage.

### 3.3. Differentiation of aMSC-Derived Neural Progenitors along the OL Lineage

Following transfer of neurospheres to adherent substratum for glial induction in cultures, cells emigrated from the spheres and formed colonies of process-bearing cells that resembled OPs (Figure 3A). Colonies of OP-like cells were isolated by cold-jet [16,17,18,19,20,21] and expanded in subcultures. At low density, OP-like cells were bi-/tri-polar in morphology (Figure 3B). As the OP-like cells approached confluence, they became polydendritic (Figure 3C). Immunocytochemistry revealed that OP-like cells were positive for the OP markers: Olig2 (Figure 3D), PDGFRα (Figure 3E), NG2 (Figure 3F), and Sox10 (Figure 3F). Flow cytometry further confirmed that >95% of the OP-like cells were positive for the four OP markers (Figure 3G–J). The results were corroborated by semiquantitative RT-PCR analysis of transcripts (Appendix A). These OP-like cells were capable of undergoing culture expansion (Appendix A, *n* = 3 individual preparations) and expressed mature OL markers CNPase and myelin basic protein (MBP) upon withdrawal of the OP mitogens bFGF and PDGF-AA [28] from the maintenance medium (Appendix A). These OP-like cells were therefore capable of maturing into OL-like cells in vitro.

### 3.4. In Vitro Myelination

The OP-like cells derived from rat aMSCs were tested for myelinating capacity in co-culture with neurons purified from embryonic day 15 rat dorsal root ganglia. After 2 weeks in co-culture, MBP-positive OLs were found with processes that associated with networks of neurofilament 200 (NF200)-positive axons (Figure 4A–C). Higher magnification images revealed individual OLs bearing MBP-positive processes alongside multiple NF200-positive axons (Figure 4D–F). Transmission electron micrographs of the MBP-positive domains revealed ultrastructure of compact myelin (Figure 4G,g*), thus confirming that these cells are functional OPs.

Cryopreserved OPs (stored under liquid nitrogen for 2 months) were similarly tested for myelinating capacity in vitro. MBP-positive OLs were observed after 2 weeks in co-culture with dorsal root ganglia neurons (Appendix A), indicating that the myelination capacity of these OPs was preserved despite cryopreservation. By contrast, neither OLs nor myelin was detectable in co-cultures of aMSCs with dorsal root ganglia neurons (Appendix A).

### 3.5. In Vivo Myelination and Extended Lifespan Follwing Transplantation of OPs

To test for the capacity of myelination in vivo, OP-like cells were transplanted into the corpus callosum of myelin-deficient shiverer mice. MBP-positive OLs were observed in the corpus callosum of recipients, both ipsilateral (Figure 5A) and contralateral (Figure 5B) to the injection site by 6 weeks post-transplantation. This indicates that OP-like cells were capable of dispersal from the injection site to other brain regions and maturation into OLs in encounters with host axons in vivo. In line with in vitro results, such OLs produced multiple MBP-positive processes in co-localization with NF200-positive axons (Figure 5a*,b*). MBP, a necessary component of compact myelin, was not detected in control shiverer mice that received vehicle injections only (Figure 6A). Bundles of myelinated axons were identified in the corpus callosum of shiverer mice by 6 weeks post-transplantation (Figure 6B) but not in control shiverer mice that received vehicle injection (Figure 6A). 

By 12 weeks post-transplantation, myelinated axons were abundant in animals that received OP transplantation (Figure 6C), but not in controls (Figure 6A). Electron micrographs further confirmed the presence of compact myelin ensheathing axons (Figure 6E,F) within the corpus callosum of shiverer mice that received OP transplant. Whereas in controls injected with vehicle, almost all OLs showed partial/no enwrapping of axons (Appendix A). A significantly higher number of axons with compact myelin were observed in recipients of OP-transplant (Figure 6G, 10.72 ± 0.83 per 100 μm² of corpus callosum) compared to recipients of vehicle (Figure 6D,G). The number of myelinated axons per 100 μm² increased to 21.16 ± 2.26 by 12 weeks post-transplantation, comparable to wild-type C57BL6J mice (Figure 6F,G). The thickness of myelin observed in recipients of OP transplant was comparable to that in wild-type controls at 12 weeks post-injection as evidenced by G-ratio (Figure 6H).

Finally, shiverer mice that received transplant of aMSC-derived OPs demonstrated significant extension of lifespan when compared to the control group (Figure 6I). From the Kaplan-Meier survival curve, the average lifespan of treated animals was significantly extended by 46.24% (Figure 6I), with an increase of the overall median survival time from 91 to 133 days. This suggested that widespread remyelination of the corpus callosum due to OP transplantation was sufficient to bring about the functional rescue of MBP-deficiency in shiverer mice.

## 4. Discussion

The current study demonstrated derivation of OPs capable of maturation into myelin-forming OLs from aMSCs. In our two-stage induction protocol, OPs were successfully derived from cryopreserved aMSCs within 20 days. Neural progenitors harbored within aMSC cultures, as shown by nestin (neural stem/progenitor marker) positivity, represent a viable cell source that could be expanded for induction into myelinating glia. The protocol presented here fosters the expansion of neuroprogenitor cells through formation of neurospheres in non-adherent culture. These progenitors are then directed to differentiate into OP-like cells in adherent culture supplemented with glia inducing factors PDGF-AA- [29] and bFGF [30]. Withdrawal of these two factors from the media triggered expression of OL markers in OPs. This mirrored the maturation trajectory of primary cultures of OPs [2,31].

A subset of neural progenitors that were positive for Olig2, the critical OL lineage determining transcription factor, was present in the neurosphere. These Olig2-positive cells likely represented progenitors poised to differentiate along the OL lineage. On the contrary, aMSC-derived Schwann cells (the myelinating glia in the peripheral nervous system) reported in our previous works [16,17,18,19,20,21] might have been obtained from Olig2-negative neural progenitors. High throughput techniques such as single-cell RNA sequencing will reveal the full differentiation potential within the neurosphere.

OP-like cells were established as functional OPs capable of maturation into myelinating OLs by co-culture with neurons, and by transplantation into shiverer mice. After 2 weeks in co-culture with neurons purified from rat dorsal root ganglia, OPs generated MBP-positive processes that co-localized with multiple axons, and assumed morphology characteristic of OLs [32]. When aMSC-derived OPs were transplanted into the brains of myelin-deficient shiverer mice. MBP-positive OLs were observed 6 weeks post-transplantation. Corroborating with the in vitro assay, MBP-positive processes of OLs were associated with NF200-postive axons. While OPs were delivered only into the right corpus callosum of homozygous shiverer mice, MBP-positive OLs were observed bilaterally in the corpus callosum by 6 weeks post-transplantation, suggesting the characteristic migration capability of OPs after transplantation [5,10]. Compact myelin with major dense lines was identified abundantly by electron microscopy when OPs encountered axons both in vitro and in vivo. Persisting survival, integration, and in vivo myelination capacity of the transplanted cells over an extensive timeframe was reflected by the increasing number of myelinated axons from week 6 to week 12. While we did not directly assess the immune response of these mice, persistent survival of xenografted cells might have been due to perinatal immune tolerance [33,34]. This is concordant to progressive myelination and axonal ensheathment by oligodendroglia over time also reported in pioneering studies [5,13].

Finally, we addressed whether myelin restoration resulted in any functional improvement. Shiverer mice that received OP transplantation had a significantly longer lifespan than non-recipients. This suggested that OPs not only mediated re-establishment of CNS myelin at a structural level but also improved neurological functions and rescued these shiverer mice from premature death due to autonomic deficits. Callosal OL networks provide energy to neurons for sustaining axonal function and action potential generation via facilitated glucose delivery [35]. This suggests the possibility of a feedback loop for activity-dependent maturation of OPs into OLs [36] and the formation of compact myelin [37,38,39]. While the limited number of OPs injected did not rescue motor deficits in shiverer mice, we postulate that future studies that focus on direct remyelination of fibre tracts in the spinal cord and sensorimotor cortices with a sufficient number of OPs would bring about the rescue of motor deficits in these mice.

Taken together, our aMSC-derived OPs were indistinguishable from OPs extracted from cortical tissue, in terms of (i) marker expression, (ii) culture conditions, (iii) in vitro maturation process, and (iv) myelin-formation capacity [27,28,31]. aMSC-derived OPs matured into myelin-forming OLs and re-established compact myelin after being transplanted into the corpus callosum of shiverer mice. Marker expression of these OPs remained stable in culture and we did not observe any tumorigenesis in mice that received OP transplant. Future transplantation studies in immune-deficient mice will ascertain the safety of aMSC-derived OPs. Treatment-dependent lifespan extension translated into the partial repair of the innate neurological and autonomic deficits in shiverer mice. This demonstrated efficacy sheds light upon future clinical applications for white matter diseases.

While transplantation of cells directly into the human brain is undoubtedly invasive, the severely debilitating symptoms of myelin disorders justify the risks of undergoing such treatments. Phase I clinical trials for injection of neural and mesenchymal stem cells into the brain of patients suffering from a variety of neurological disorders have been conducted globally with no adverse effects reported in longitudinal studies [8,9,10]. Furthermore, intrathecal injection also presents itself as a lower-risk option for the introduction of cells into the CNS [40]. With these trials demonstrating the safety of intracranial cell transplantation, clinical use of OP transplant is only limited by the lack of a lineage stable and easy to obtain cell source. Bone marrow aspiration from the posterior iliac crest is an established procedure for obtaining bone marrow [41,42]. Given that aMSC-derived OPs can be culture-expanded, only a limited amount of bone marrow would be sufficient to produce OPs on the scale required for therapeutic use. With lineage stability after cryopreservation and retention of myelin-forming capacity following reconstitution, OPs thus derived have the potential to become “off-the-shelf” products. These advantages serve as a solid foundation for future translations of industrial implications. 

## 5. Conclusions

Prior to our study, reported sources of OPs mainly resort to human embryonic stem cells [5,7], iPSCs [13,14,15], or through direct reprogramming of fibroblast [32]. Despite being effective in remyelination, these approaches confront either ethical issues or safety/technical challenges. Through adopting aMSCs, we avoided ethical issues of using human fetal tissue or embryonic stem cells [11,12]. We demonstrated that aMSC cultures harbored neural progenitors that can be directed to differentiate along the OL lineage. Through manipulation of the in vitro niche, including the culture substratum and a combination of growth factors, we were able to derive OPs from cryopreserved aMSCs without genetic/epigenetic manipulation or viral vectors. This circumvents a major technical/safety challenge faced by direct reprogramming [32]. OPs thus generated were capable of maturation into OLs that form compact myelin both in vitro and in vivo. Transplantation of these OPs restored compact myelin in the corpus callosum of myelin-deficient shiverer mice and significantly extended their lifespan. These findings hold promise for establishing cell therapies in the CNS tackling both congenital and acquired myelin disorders.

## Figures and Tables

**Figure 1 cells-10-02166-f001:**
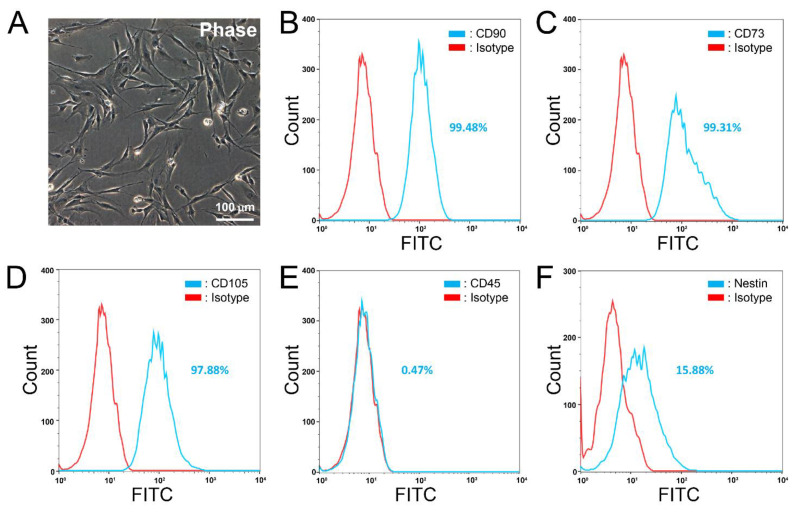
Flow cytometric analysis of aMSCs in culture. (**A**) Phase contrast micrograph showing fibroblastic morphology typical of aMSCs in passages 3–5. (**B**–**D**) Representative histograms showing high expression of mesenchymal stem cell markers CD90 (**B**, 99.48%), CD73 (**C**, 99.31%) and CD105 (**D**, 97.88%). Hematopoietic marker CD45 (**E**) was only positive in 0.47% of cells. The neural progenitor marker nestin was positive in 15.88% of cells (**F**). *n* = 4 independent preparations of aMSCs.

**Figure 2 cells-10-02166-f002:**
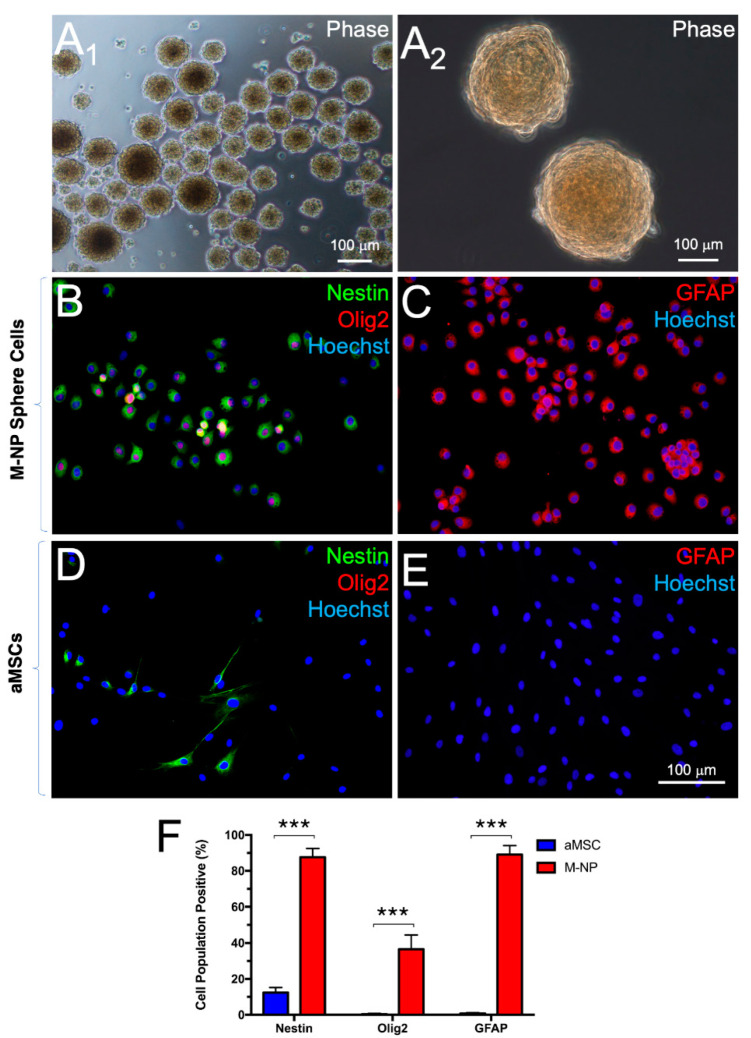
Selective expansion of neural progenitors in non-adherent culture. aMSCs formed spheres by day 6 in non-adherent culture (**A_1,2_**). Increased expression of nestin (**B**), Olig2 (**B**), and GFAP (**C**) were observed in the course of non-adherent culture versus parallel, adherent cultures of aMSCs (**D**,**E**). Histogram showing the percentage of aMSCs positive for the indicated markers (**F**). *n* = 5 independent preparations. ***: *p* < 0.0001. Student’s *t*-test.

**Figure 3 cells-10-02166-f003:**
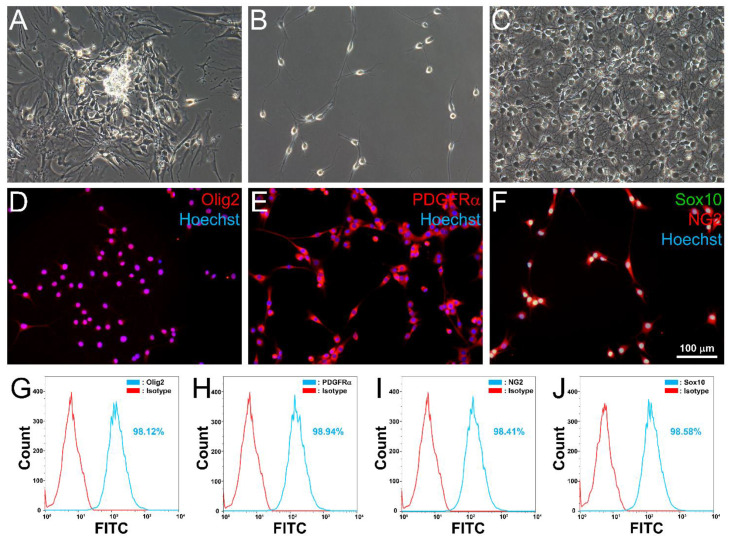
Derivation of OP-like cells. Spindle-shaped cells exited from adherent spheres 7 days after glia induction in adherent culture (**A**). Following re-plating, cells developed bi/tri-polar OP-like morphology (**B**). At higher cell density, OP-like cells became polydendritic (**C**). These cells were positive for the OP markers Olig2 (**D**), PDGFRα (**E**), NG2, and Sox10 (**F**). Flow cytometric analysis of OP-like cells found then immunopositive for Olig2, PDGFRα, NG2, and Sox10 in 98.12% (**G**), 98.94% (**H**), 98.41% (**I**), and 98.56% (**J**) of cells respectively. Histograms of a representative experiment are shown. *n* = 4 independent preparations.

**Figure 4 cells-10-02166-f004:**
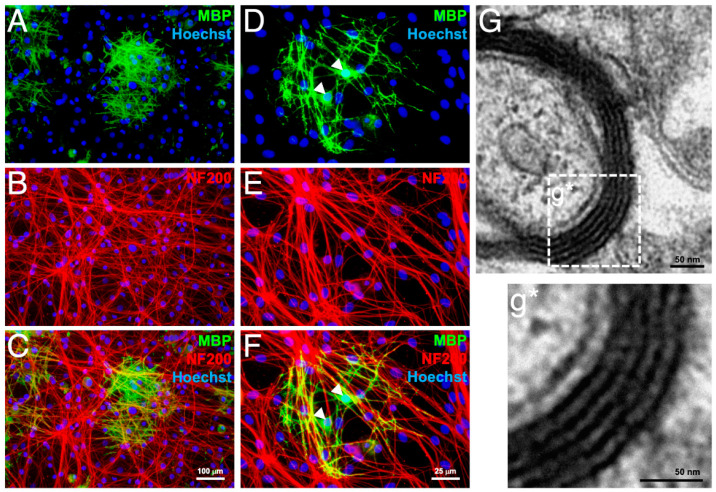
Maturation of OP-like cells into myelinating OLs in co-culture with neurons purified from embryonic day 15 rat dorsal root ganglia. After 2 weeks in co-culture, MBP-positive mature OLs (**A**) extended processes along NF200-positive axons (**B**, merged image in **C**). Higher magnification revealed individual OLs bearing multiple MBP-positive processes (**D**–**F**). OL cell bodies are indicated by white arrowheads (**D**,**F**). Transmission electron micrograph showing ultrastructure of compact myelin in a sample taken from an MBP-positive domain in the co-culture (**G**,**g***). *n* = 4 independent preparations.

**Figure 5 cells-10-02166-f005:**
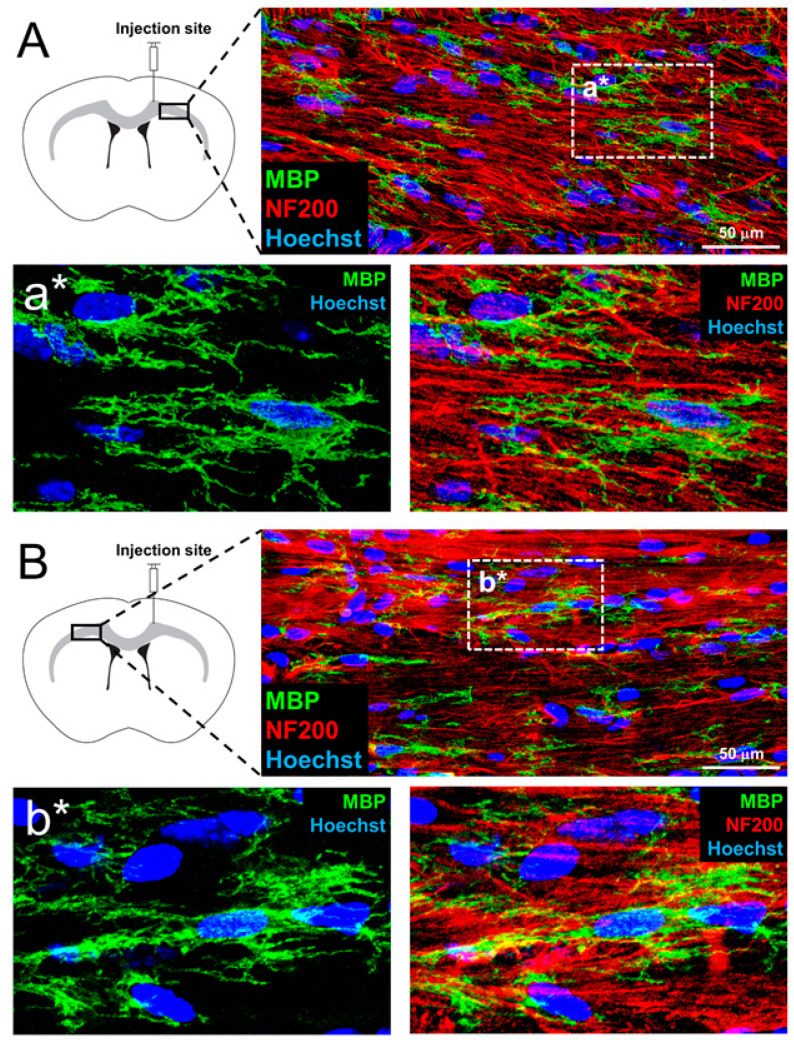
Migration and maturation of aMSC-derived OPs into myelinating OLs in myelin-deficient shiverer mouse model, 6 weeks post-transplantation into the corpus callosum was performed at the right hemisphere. MBP-positive OLs were observed in the corpus callosum of, on both the right (**A**) and the left (**B**) 6 weeks post-transplantation. Multiple MBP-positive processes along NF200-positive axons (**a***,**b***). *n* = 5 mice.

**Figure 6 cells-10-02166-f006:**
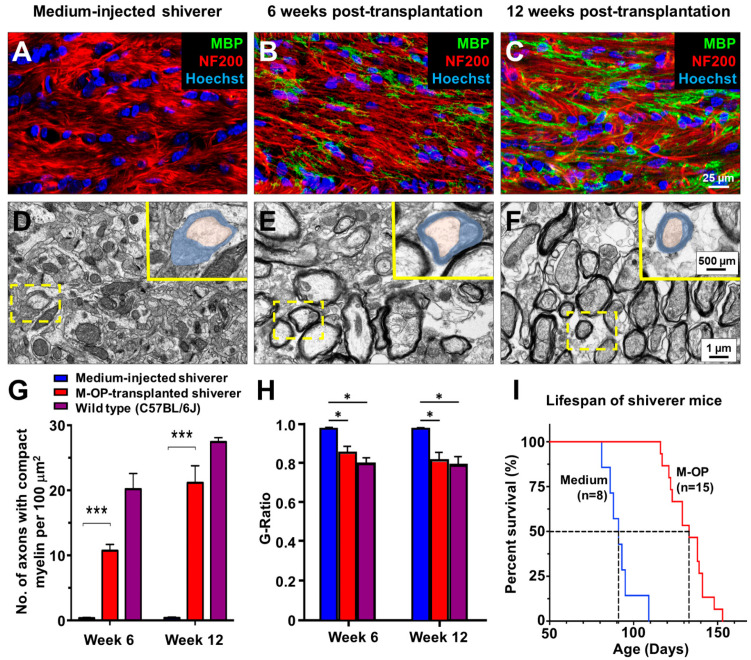
Compact myelin in the corpus callosum and lifespan extension of shiverer mice that received OP transplant into the corpus callosum. Medium-injected controls display no MBP-positivity in the corpus callosum (**A**). Recipients of OP transplants displayed MBP-positivity in the corpus 6 (**B**) and 12 weeks (**C**) post-transplantation. Transmission electron micrographs showing compact myelin with major dense lines in OP recipients 6- (**E**) and 12-weeks (**F**) post-transplantation, but not in the medium-injected controls (**D**) by week 12. Insets highlight OLs (blue) enwrapping axons (orange) (**D**–**F**). The density of axons enwrapped by compact myelin was significantly higher (**G**) with G-ratios comparable to that of wild-type mice (**H**) in OP recipients compared to controls. OP recipient shiverer mice: *n* = 5 (6-week) and *n* = 4 (12-week), control shiverer mice: *n* = 4 (for both time points). C57BL/6J mice: *n* = 4 (for both time points). The average lifespan of the OP recipients was significantly extended by 46.24% (**I**), with a median survival time of 133 days compared to 91 days in controls. *: *p* < 0.05, ***: *p* < 0.0001. Student’s *t*-test. Shiverer mice: *n* = 15 OP recipients, *n* = 8 medium controls.

## Data Availability

Datasets of the current study are available from the corresponding authors upon request.

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
