# Peer review of "Derivation of Oligodendrocyte Precursors from Adult Bone Marrow Stromal Cells for Remyelination Therapy"

_cells, 2021, doi:10.3390/cells10082166_

Round 1

Reviewer 1 Report

In this work the authors developed a protocol for isolation and differentiation of oligodendrocyte precursor cells (OPCs) from adult mesenchymal stem cells (aMSC) and tested their function in an in-vitro and in-vivo environments. After the isolation of the OPCs, they were divided into two experiments. In the first experiment they were placed in a co-culture with a pure neuronal culture from the dorsal root ganglion in order to examine whether the OPCs can mature into myelinating oligodendrocytes that envelop neurons’ axons. In the second experiment they were transplanted into a shiverer mouse model’s brain in order to examine their function, migration and their impact on lifespan. The results showed that OPCs that are derived from aMSCs can mature into myelinating oligodendrocytes that interact with neurons both in-vitro and in-vivo, and prolong the lifespan of the shiverer mice compared to the control group. The results were observed using various methods, including electron microscopy in order to examine the thickness of the myelin layer that was formed around the axons, immunohistochemistry and immunocytochemistry in order to see the expression of myelin related genes, and real-time PCR.

In my opinion, this work touches upon an important subject as there are many neurological conditions that are associated with myelin deficiency and this kind of treatment might ameliorate some symptoms in those conditions. Their protocol suggests a new source for these cells that uses a greatly simplified isolation and differentiation process of the oligodendrocytes compared to previous methods (e.g. iPSCs), which is important as the source of such cells is very limited. Another interesting aspect of this work is the use of cryopreserved aMSC and OPCs and the examination of its impact on cell viability. This procedure could be important to future clinical treatment as it could be preserved for some time and not affect the efficacy of the treatment.

However, I have some concerns regarding several issues in the paper, and the writing should be improved in several parts:

-I was surprised by the claim that the amount of myelinated axons in the control shiverer mice is zero. It is reasonable that the control group will have fewer myelinated axons, and that the myelin would be less compact (lamellae etc.), but lacking myelin completely sounds odd. This matter should be further examined, including a performance of immunohistochemical staining using a variety of proteins and not only Mbp, and if possible discuss prior works on this mouse model that support this claim.

-The introduction lacks many previous works and lacks a detailed overview of aMSCs, e.g. a general description of those cells, the harvesting process, its uses, and its clinical applicability for different pathologies. In addition, there are some methods that are described, but are missing in the results section. Several other topics also have lacking overviews, e.g. cryopreservation.

-There is a large emphasis in the paper on the treatment’s clinical application, however this is a very invasive procedure (injection to the brain) - this should be discussed and solutions should be offered for that matter.

-In the discussion, it is mentioned that there wasn’t an immune reaction to the transplant, but no results were shown in the result section to support that claim. Authors should either show empirical evidence for that claim, or remove it. 

- Only the improvement of lifespan was examined in this work. The impact on behavioral phenotypes related to myelin deficiency other than lifespan, such as motor skills, should be further investigated, to support the advantages of the treatment.  

Minor comments:

Line 15: the context of the sentence is not clear, and needs to be combined with the earlier sentence.

Line 34: consider changing the word “theme”

Line 39: consider adding more disadvantages to the use of iPSCs (teratomas that could occur after transplantation, can it also occur in aMSC?)

Lines 47- 49: rephrase the sentence. By “values” do you mean “evaluate”? What do you mean by “readiness”?

Lines 49-50: rephrase the sentence.  Change the beginning of the sentence (“off the shelf”) and give a more detailed overview of cryopreservation and its significance to clinical trials. 

Lines 52-58: consider moving this paragraph in the methods section. It seems very in-depth instead of a general description of the procedure.

Line 101: need to add space between the words “and”, “the”.

Line 107: - “except cells” -> “, except that the cells”

Line 168: the origin of the OPCs that were injected to the shiverer mice were taken originally from mice or rats? It’s unclear.

Line 172: You state that in total 42 mice were used. How many mice were in the experimental group and how many in the control group?

Line 179: consider separating immunohistochemical staining and immunocytochemical staining in the method section.

Line 289: consider moving figure 4 to after line 309.

Line 299: coculture -> co-culture

Line 310: consider moving figure 5, 6 to after line 357.

Line 333: it would be advisable to label the OPCs that were transplanted in order to be certain that the cells you examine are the oligodendrocytes that were transplanted and not ones that were in the mice brain from before.

Line 337: It seems that your results indicate a complete lack of white matter, and not only a deficiency. This results seems a bit odd (see note in my major comments).

Line 407: you mention that there wasn’t a reaction of the immune system to the OPCs that were injected, but I could not find these results.

Line 452: discuss disadvantages to this method, how you can overcome this issues, and how it could be applied to humans as well. What is the procedure in taking bone marrow from humans? Is it possible to take it from a person with the pathology or would it affect the development of the transplanted oligodendrocytes? If the procedure includes direct injection to the brain, it still has clinical feasibility? Was a similar procedure performed in humans using other type of cells in different pathologies?

Reviewer 2 Report

Critique:

The manuscript presents interesting and timely data and is generally clearly written and documented. The investigators developed a rapid method for generating oligodendrocyte precursor cells from stored adult mesenchymal cells isolated from rat femur bone marrow and after expansion and differentiation showed that the cells myelinated dorsal root ganglion cells grown in culture and fiber tracks in myelin-lacking shiverer mice where normal myelination does not occur. 

I would have liked to have a little more information on the strategy the investigators used in choosing factors to first expand neural progenitors in the mesenchymal cells and then to differentiate the cells into OPs. Were conditions varied and optimized? Are there possibilities for increasing expansion rates and or differentiation? Are there possibilities that small changes could compromise the approach?

I would also like a bit more follow up on the longevity of the shiverer treated with M-OPs. As other regions, including heavily myelinated spinal cord were likely still without myelin, did the investigators see movement and/or sensory deficits that would occur with unmyelinated spinal cord circuits? Was any myelination seen outside the corpus callosum, e.g., in the cortex, cerebellum?

As this study using rodent mesenchymal cells for the rescue in mice and cell culture, some thought could be forwarded in the discussion on generating human M-OPs and using them for treatment in humans.

Minor points: The overall writing could be improved for English. If the investigators could find a person to edit the English it might help keep readers more engaged, though honestly, the work speaks for itself.

I am providing a few examples for improvements of English in the abstract –

  1. Although OPs and M-OPs were introduced as abbreviations in the abstract, M-NPs should have been included (line 16).
  2. Over should replace across in line 22.
  3. Becoming comparable to that of would sound better being replaced by approaching levels in wild type (line 23).
